Exploring the potential biomarkers for prognosis of glioblastoma via weighted gene co-expression network analysis

Zhang Mengyuan 1
Zhou Zhike 2
Liu Zhouyang 1
Liu Fangxi 1
Zhao Chuansheng cszhao@cmu.edu.cn 1
1 Department of Neurology and Stroke Center, The First Hospital of China Medical University , Shenyang , China
2 Department of Geriatrics, The First Hospital of China Medical University , Shenyang , China
Shang Yuan
Electronic publication date: 2022 Jan 18
Publication date: 2022
Volume: 10
Electronic Location ID: e12768
Received 2021 Jun 11; Accepted 2021 Dec 17
Copyright: ©2022 Zhang et al.
Copyright year: 2022
Copyright holder: Zhang et al.
License: This is an open access article distributed under the terms of the Creative Commons Attribution License, which permits unrestricted use, distribution, reproduction and adaptation in any medium and for any purpose provided that it is properly attributed. For attribution, the original author(s), title, publication source (PeerJ) and either DOI or URL of the article must be cited.
License URL: https://creativecommons.org/licenses/by/4.0/

Keywords: Glioblastoma, Biomarkers, Prognosis, Weighted gene co-expression network analysis, GEO

Funding: The Program for Liaoning Innovative Research Team in University LT2019015 LiaoNing Revitalization Talents Program and Liaoning Key R&D Guidance Program 2019JH8/10300002 This study was supported by the Program for Liaoning Innovative Research Team in University (NO.LT2019015) and the LiaoNing Revitalization Talents Program and Liaoning Key R&D Guidance Program (NO. 2019JH8/10300002). The funders had no role in study design, data collection and analysis, decision to publish, or preparation of the manuscript.

==============================
Background

Glioblastoma (GBM) is the most common malignant tumor in the central system with a poor prognosis. Due to the complexity of its molecular mechanism, the recurrence rate and mortality rate of GBM patients are still high. Therefore, there is an urgent need to screen GBM biomarkers to prove the therapeutic effect and improve the prognosis.

Results

We extracted data from GBM patients from the Gene Expression Integration Database (GEO), analyzed differentially expressed genes in GEO and identified key modules by weighted gene co-expression network analysis (WGCNA). GSE145128 data was obtained from the GEO database, and the darkturquoise module was determined to be the most relevant to the GBM prognosis by WGCNA (r =  − 0.62, p = 0.01). We performed enrichment analysis of Gene Ontology (GO) and Kyoto Encyclopedia of Genes and Genomes (KEGG) to reveal the interaction activity in the selected modules. Then Kaplan-Meier survival curve analysis was used to extract genes closely related to GBM prognosis. We used Kaplan-Meier survival curves to analyze the 139 genes in the darkturquoise module, identified four genes (DARS/GDI2/P4HA2/TRUB1) associated with prognostic GBM. Low expression of DARS/GDI2/TRUB1 and high expression of P4HA2 had a poor prognosis. Finally, we used tumor genome map (TCGA) data, verified the characteristics of hub genes through Co-expression analysis, Drug sensitivity analysis, TIMER database analysis and GSVA analysis. We downloaded the data of GBM from the TCGA database, the results of co-expression analysis showed that DARS/GDI2/P4HA2/TRUB1 could regulate the development of GBM by affecting genes such as CDC73/CDC123/B4GALT1/CUL2. Drug sensitivity analysis showed that genes are involved in many classic Cancer-related pathways including TSC/mTOR, RAS/MAPK.TIMER database analysis showed DARS expression is positively correlated with tumor purity (cor = 0.125, p = 1.07e−02)), P4HA2 expression is negatively correlated with tumor purity (cor =−0.279, p = 6.06e−09). Finally, GSVA analysis found that DARS/GDI2/P4HA2/TRUB1 gene sets are closely related to the occurrence of cancer.

Conclusion

We used two public databases to identify four valuable biomarkers for GBM prognosis, namely DARS/GDI2/P4HA2/TRUB1, which have potential clinical application value and can be used as prognostic markers for GBM.

Introduction

Glioblastoma (GBM) is the most common primary neurogenic tumor, and the prognosis of most subtypes is poor (Tan et al., 2020). Despite of aggressive treatment strategies such as surgery followed by irradiation and chemotherapy, the prognoses of GBM patients remained unsatisfactory (Wu et al., 2020). According to the existing data, GBM patients have a survival of only 12–15 months after the standard treatment, with the 5-year survival rate of 3–5% (Gong et al., 2020; Szopa et al., 2017). The main reasons for the poor prognosis of GBM are due to tumor metastasis and postoperative recurrence (Tij et al., 2021). Given that tumors invade the brain aggressively, GBM tumors can rarely be completely removed by surgery (Reichel et al., 2020). The resulting network by GBM enables multicellular communication through microtube-associated gap junctions, and increases tumor resistance to cell ablation and radiotherapy (Li et al., 2017a). Actively searching for biological markers related to the treatment and prognosis of GBM patients is of great significance for improving the survival rate of GBM patients.

In the past few decades, gene sequencing and bioinformatics analysis have been widely used for genetic variation screening at the gene level (Tingting et al., 2019), which helps us to identify differentially expressed genes (DEG) and functional pathways in the development of GBM. It has been found that the increased expression of SPRY2 mRNA indicates the decreased survival rate of GBM patients (Li et al., 2017a). Another study showed that the mRNA levels of NOTCH and Epidermal Growth Factor Receptor (EGFR) genes were increased in GBM tissues, which was related to the survival of patients (Irshad et al., 2015; Xing, Sun & Guo, 2015). However, most of these studies are single gene analysis, which may limit the analysis of the pathogenesis and prognosis of GBM.

Weighted gene co-expression network analysis (WGCNA) is a platform to identify hub genes or therapeutic targets based on the interconnectivity of gene subsets and the association between gene subsets and phenotypes (Wang et al., 2020b; Zhang & Horvath, 2005). WGCNA can use the information of thousands of genes to identify the gene modules of interest and perform important association analysis on phenotypes. Recently, many journals have published relevant studies using WGCNA (Schafer et al., 2019; Wang et al., 2020b; Zhou et al., 2018).

In this study, we extracted four GBM related biomarkers (DARS / GDI2/P4HA2/TRUB1) by extracting data from GBM patients from the gene expression integrated database (GEO) and using WGCNA and Kaplan–Meier survival curves analysis. Then, we established GBM gene markers in the tumor genome atlas (TCGA), and confirmed the characteristics of these four genes by means of co-expression analysis, drug sensitivity analysis, TIMER database analysis, and GSVA analysis. In summary, our purpose is to find reliable biomarkers related to the prognosis of GBM by analyzing the relationship between DARS/GDI2/P4HA2/TRUB1 gene and GBM, so as to provide reference and basis for clinical treatment and prognosis observation of GBM.

Material and Methods

Data information and construction of WGCNA

The Series Matrix file data File of GSE145128 was downloaded from the NCBI GEO public database, which were contained 15 GBM patients and sets of transcriptional data, including untreated group (n = 7) and recurrent group (n = 8), for the construction of WGCNA co-expression network.

We constructed a weighted gene co-expression network to find co-expressed gene modules, and clarified the relationship between the gene network and phenotype and hub genes. The WGCNA-R package was used to construct a co-expression network of genes in the GSE145128 dataset, where the soft-thresholding power was set to 16. The weighted adjacency matrix is converted into a topological overlap matrix (TOM) to estimate the network connectivity, and a hierarchical clustering method is used to construct a clustering tree structure of the TOM matrix. Different gene modules are represented by different cluster tree branches and colors. All genes are divided into multiple modules through gene expression patterns, and genes with similar expression patterns are divided into one module based on weighted correlation coefficients and expression patterns.

Enrichment analysis of gene module function

In order to obtain the biological functions and signaling pathways involved in the interest module of WGCNA, the Metascape database (http://www.metascape.org) was used for annotation and visualization, and Gene Ontology (GO) analysis and Kyoto Encyclopedia of Genes and Genomes (KEGG) pathway enrichment analyses were performed on the genes in the specific module. Min overlap ≥3 & p ≤ 0.01 was considered statistically significant.

Identifying key genes

To determine key genes, the most important thing is whether they have an impact on tumor prognosis. According to WGCNA theory, key genes have the highest connectivity in the module, which determines the biological significance of the module (Chen et al., 2019). So we think that key genes must exist in the interested module of WGCNA. Combined with the above two points, we analyzed the Kaplan–Meier survival rate of all genes in the interest module of WGCNA. We believe that the genes that can affect the prognosis of GBM patients are the key genes. And then,the next step is to explore and verify the specific molecular mechanism of key genes.

Download and pre-processing data from TCGA

TCGA database as the biggest cancer gene information database, including gene expression data, the miRNA expression data and copy number variation, DNA methylation, SNPS and other data. We downloaded the processed original mRNA expression data of GBM. A total of 159 specimens were collected (Table S1).

Co-expression analysis

The co-expression of the key genes were analyzed. The correlation coefficient filter condition was 0.3 and the p-value was 0.001. After screening the genes with the most significant expression of key genes, the correlation analysis circles of key genes were plotted using “corrplot” and “circlize” packages.

GSCALite and GDSC

GSCALite is a set analysis platform for cancer genes. It integrates cancer genomics data of 33 cancer types from TCGA, drug response data from GDSC and CTRP, and normal tissue data from GTEx, and conducts gene set analysis in the data analysis process. Our study through this analysis was carried out on the key genes.Then base on the largest publicly available pharmacogenomics database (GDSC, the Genomics of Drug Sensitivity in Cancer, https://www.cancerrxgene.org/), we used the R packet “pRophetic” to predict the chemosensitivity of each tumor sample, and the estimated IC_ (50) of each specific chemotherapeutic drug treatment was obtained by ridge regression. The prediction accuracy was measured by 10-fold cross-validation with the GDSC training set. Select default values for all parameters, including “combats” to remove batch effects, “allSoldTumours” for tissue types, and average values for summarizing repetitive gene expressions (Liu et al., 2018).

TIMER database analysis

TIMER is a website for systematically testing the molecular characteristics of tumor-immune interactions (Li et al., 2017b). This website has incorporated 10,897 samples ranging from 32 different kinds of cancer types from the TCGA dataset (Shi et al., 2020). In this study, TIMER was used to explore the relationship between key genes and the contents of immune cells and to compare the infiltration levels between tumors with different somatic copy number changes of key genes.

Gene functional analysis

GSVA uses a non-parametric and unsupervised method, and bypasses the traditional method of explicitly modeling phenotypes in affluent scoring algorithms (Hanzelmann, Castelo & Guinney, 2013). By comprehensively scoring the gene set of interest, GSVA converts the gene level change into the pathway level change, and then judges the biological function of the sample. In this study, the gene sets were downloaded from the Molecular signatures database (v7.0 version),and used the algorithm of GSVA comprehensive score of each gene set, evaluating the potential biological function change different samples.

Statistical analysis

All statistical analyses were performed in R language (version 3.6). All the statistical tests were bilateral, and p < 0.05 was statistically significant.

Result

Identification of gene co-expression modules

The Series Matrix File data File of GSE145128 was downloaded from the NCBI GEO public database. A total of 15 transcriptional data sets, including untreated group (n = 7) and recurrent group (n = 8), were used to construct the WGCNA co-expression network. In order to determine whether the 15 samples in GSE145128 were suitable for network analysis, a sample dendrogram and similar clinical features were studied. We confirmed that all samples were included in the group (Fig. 1A). The soft-thresholding power was set as 16 for the subsequent construction of co-expression network (Figs. 1B, 1C). The clustering tree structure of TOM matrix was constructed by hierarchical clustering method. The different branches and colors represent different gene modules (Fig. 1D). The network heatmap was used to analyze the interaction of 41 modules (Fig. 1E). The results showed that each module was independent of each other, indicating that each module was highly personalized and the gene expression of each module was relatively independent.

Figure 1 Identification of gene co-expression modules.

(A) Cluster samples to detect outliers. All samples are located in the cluster and pass the critical threshold at the same time. The green highlighting means that the samples are in strong trait relationships established by correlation analyses. (B) The scale-free fit index was analyzed under the background of different soft-thresholding power (β). (C) Analyze average connectivity when using different soft-thresholding powers. (D) Dendrogram clustering of all genomic genes in GBM samples. (E) Heatmap of co-expressed genes. Different modules on the X and Y axis have different colors. The connection degree of different modules is indicated by the yellow intensity.

Correlation of modules and clinical traits

In order to study the relationship between these modules and the prognosis of GBM, we investigated the correlation between each module and the prognosis of GBM.We found that the darkturquoise module had the highest correlation with disease relapse (r =  − 0.62, p = 0.01); (Figs. 2A, 2B). We used Metascape to analyze the function and pathway of the darkturquoise module. Metascape can identify the enrichment process in the gene list and the association between enrichment processes (Tripathi et al., 2015a; Shannon et al., 2003) by querying many databases,such as GO functional, Hallmark Gene Sets, and KEGG pathways (Tripathi et al., 2015; Zheng et al., 2018). Based on GO enrichment analysis, it was found that the co-expressed genes within the modules of interest s mainly related to the steady state of cellular transition metal ion homeostasis, protein hydroxylation, intrinsic apoptotic signaling pathway, ncRNA metabolic process (Fig. 2C). The KEGG pathway analysis revealed that the co-expressed genes within the modules of interest was mostly enriched in the ‘Ferroptosis’ (Fig. 2C). In addition, the enrichment processes were highly connected and could be clustered into a complete network (Fig. 2C). These results indicated that these functions were related in the occurrence and development of GBM.

Figure 2 Correlation of modules and clinical traits.

(A) Module intrinsic genes and relapse Heatmap of the correlation between. (B) Scatter plot of the correlation between the darkturquoise module and relapse. All modules can be correlated with genes, and all continuous traits can be correlated with gene expression values. The two correlation matrices are combined and the vertical coordinate is the Gene significance for luminal when the module of interest is specified for analysis. (C) Enrichment analysis of the function and pathway of the darkturquoise module. The rich biological process terms in the selected modules are described as interactive networks and listed according to their P-value. The size of the dots represents the number of genes that are co-expressed, the larger the dot, the more genes are co-expressed, presumably the more important they are and the more important they are as core genes in the network graph. Each node is a gene. The size of the node means degree of gene enrichment. Set P < 0.01 as the cutoff criterion. Enrich the term network, colored with cluster-ID, where nodes sharing the same cluster ID are usually close to each other.

Identification of key genes in darkturquoise module

According to the WGCNA theory, key genes have the highest connectivity in the module, which determine the biological significance of the module and therefore influence the survial of patients intensively (Chen et al., 2019). Therefore, we searched for hub genes in the darkturquoise module. We analyzed the Kaplan–Meier survival curves of 139 genes in the darkturquoise module. Fourteen genes with significant survival analysis results (p < 0.05) were selected for sequencing (Table 1). Further determination of the key gene requires a combination of its expression, typicality in previous studies, and previous research experience in our lab. Among the 14 candidate genes we selected, most genes have been confirmed to be related to the prognosis of GBM, such as: SPAG4, FKBP1B, DLEU1, PRKAR2B, NRL, CD24, GAS6; some genes have not been clearly studied to be related to the occurrence of any known tumors, such as CAMSAP2; the remaining genes have not been significantly expressed in GBM, such as CORO6 (Lo et al., 2009; Sun et al., 2021; Wang et al., 2019; Zhao et al., 2019). Finally, we found that four genes (DARS/GDI2/P4HA2/TRUB1) had an impact on the survival rate of patients with GBM, and were also confirmed to be related to tumorigenesis. At the same time, they were not confirmed to participate in the occurrence and development of GBM, which met the conditions for our further research (Fig. 3). Survival analysis showed that the patients with low expression of DARS/GDI2/TRUB1 and high expression of P4HA2 had poor prognosis.

Analysis of the co-expression of key genes

It was clear that the key genes can affect the process of disease progression by regulating related genes. It can be assumed that DARS/GDI2/P4HA2/TRUB1 was associated with the most abundant pathways and genes and could regulate more biological processes. In order to assess the gene correlation of DARS/GDI2/P4HA2/TRUB1, we analyzed the co-expression of DARS/GDI2/P4HA2/TRUB1 through Pearson correlation analysis(cor>0.3, p < 0.001). We screened the 10 genes with the strongest correlation with the expression of DARS/GDI2/P4HA2/TRUB1, drew the correlation analysis map and heat map of DARS/GDI2/P4HA2/TRUB1 (Figs. 4A–4H), and found that the correlation between DARS and CDC73 was the highest, and the correlation between GDI2 and CDC123 was the highest. P4HA2 and B4GALT1 have the highest correlation, TRUB1 and CUL2 have the highest correlation (Figs. 4I–4L). Then,we verified the modules of these four genes in WGCNA. We found that CDC73, CDC123 and CUL2 all exist in the darkturquoise module, which is consistent with DARS/GDI2/P4HA2/TRUB1 (Table S2). Although B4GALT1 does not exist in the darkturquoise module, studies have confirmed that B4GALT1 can affect the development of GBM by regulating the apoptosis and autophagy (Wang, Li & Xie, 2020a). Among them, CDC73 and CDC123 are cyclins of cell division (Sun et al., 2017), B4GALT1 is one of seven β - 1, 4-galactosyltransferases (B4GALT). CUL2 contributes to form E3 ubiquitin ligase that can recognize numerous substrates and is involved in a variety of cellular processes (Liu et al., 2019). These four genes have been shown to have a close relationship with many kinds of cancers (Cao et al., 2020a; Dou et al., 2020; Li et al., 2019a), such as thyroid carcinoma (Sarquis et al., 2019), breast cancer, etc.

Table 1 Statistics of genes in darkturquoise modules.

Gene	P value	
TRUB1	2.25E−02	
P4HA2	1.02E−02	
DARS	4.27E−02	
FKBP1B	6.13E−03	
NRL	2.20E−02	
CORO6	1.83E−02	
LRRC43	4.65E−02	
GAS6	3.63E−02	
SPAG4	2.07E−03	
PRKAR2B	1.48E−02	
CAMSAP2	1.31E−02	
CD24	2.52E−02	
GDI2	4.42E−03	
DLEU1	1.45E−02	

Figure 3 The Kaplan–Meier survival curve can evaluate the prognostic performance of core genes based on the expression status of selected biomarkers in the database.

(A) DARS. (B) GDI2. (C) P4HA2. (D)TRUB1. All patients in each group were divided into high expression group and low expression group by gene expression. The cutoff for low versus high expression is 3-fold expression of controls.

Figure 4 Gene co-expression.

(A–H) In the TCGA dataset, selected the mRNA expression levels of DARS/GDI2/P4HA2/TRUB1 related genes, analyzed the correlation of these genes through R, and visualize them with the circus and heatmap graph. (I–L) The four genes with the highest correlation with DARS/GDI2/P4HA2/TRUB1, drew scatter plots.

Cancer-related pathways and drug sensitivity analysis of key genes

First, we investigated the role of key genes in all well-known cancer-related pathways, as the following: TSC/mTOR, RTK, RAS/MAPK, PI3K/AKT, Hormone ER, Hormone AR, EMT, DNA Damage Response, Cell Cycle, Apoptosis pathways. The results found that DARS participated in the TSC/mTOR pathway activation; GDI2 was involved in Apoptosis, TSC/mTOR pathway activation; P4HA2 was involved in DNA Damage Response, EMT, Hormone AR, Hormone ER, RAS/MAPK and TSC/mTOR pathway; TRUB1 was involved in Apoptosis, DNA Damage Response, EMT, Hormone AR, PI3K/AKT and TSC/mTOR pathway (Fig. 5A). To investigate whether the expression of DARS/GDI2/P4HA2/TRUB1 in GBM had an impact on treatment (e.g., chemotherapies), we constructed a predictive model on six commonly used chemo drugs (i.e., AKT.inhibitor, Cisplatin, Dasatinib, Erlotinib, Gefitinib, and Gemcitabine) and confirmed that high expression of DARS was less sensitive to Cisplatin (p = 0.00026) and Gemcitabine (p = 0.0024), high expression of GDI2 was less sensitive to AKT.inhibitor (p = 5.2e −05), Cisplatin (p = 0.00067), Dasatinib (p = 0.012) and Gemcitabine (p = 3.5e −06), low expression of P4HA2 was less sensitive to Cisplatin (p = 0.0032), and Gemcitabine (p = 0.00017), and high expression of TRUB1 was less sensitive to AKT. inhibitor (p = 0.00043)(Fig. 5B).

Figure 5 Drug sensitivity analysis.

(A) The role of DARS/GDI2/P4HA2/TRUB1 in the famous cancer related pathways (GSCALite). The size of an area in the pie chart represents the extent of the role of DARS/GDI2/P4HA2/TRUB1 in the well-known cancer-related pathway (GSCALite). (B) In the GDSC training set, high expression of DARS/GDI2/P4HA2/TRUB1 was inferred to be less sensitive to commonly used chemotherapy drugs. The test for association between paired samples used Pearson’s correlation coefficient. Two-tailed statistical P values were calculated by a two-sample Mann–Whitney test or Student’s t test when appropriate.

Immune cells infiltration analysis

In view of the obvious prognostic value of DARS/GDI2/P4HA2/TRUB1 gene, we used the TIMER database to determine whether there was an association between tumor-infiltrating and immune cells and DARS/GDI2/P4HA2/TRUB1 expression.Results showed that DARS expression was positively correlated with tumor purity, P4HA2 and B cells (partial cor =−0.239, p = 7.89e−07), P4HA2 and CD8+ T cells (partial cor =−0.158, p = 1.19e−03), TRUB1 and CD8+ T cells (partial cor =−0.206, p = 1.87e−02), DARS and neutrophils (partial cor = 0.245, p = 3.73e−07), GDI2 and neutrophils (partial cor =0.184, p = 1.62e−04), GDI2 and Dendritic cells (partial cor =0.167, p = 6.21e−04) P4HA2 and B cells (partial cor =−0.239, p = 7.89e−07), P4HA2 and CD8+ T cells (partial cor =−0.158, p = 1.19e−03), TRUB1 and CD8+ T cells (partial cor =−0.206, p = 1.87e−02), DARS and neutrophils (partial cor = 0.245, p = 3.73e−07), GDI2 and neutrophils (partial cor =0.184, p = 1.62e−04), GDI2 and Dendritic cells (partial cor =0.167, p = 6.21e−04) (Fig. 6A). We also explored the correlation between tumor immune cell infiltration and somatic copy number alterations (SCNAs). The samples were divided into four types according to the copy number of genes.The distribution of infiltrating immune cells among the four types of samples was compared, as shown in Fig. 6B. We found that the various forms of mutations carried by the DARS/GDI2/P4HA2 / TRUB1 gene can usually suppress immune infiltration, including CD8+T cells, neutrophils, dendritic cells, macrophages, CD4+T cells, and B cells. Also, we found that these four pivotal genes had a greater effect on immune infiltration than alterations in the genes.

Figure 6 Genetic and transcriptional changes and connections with immune cell populations.

(A) Correlation of DARS/GDI2/P4HA2/TRUB1 expression with immune infiltration level in GBM. (B) DARS/GDI2/P4HA2/TRUB1 copy number alterations (CNV) affects the level of infiltration of B cells, CD8+ T cells, CD4+T celles, Macrophages, Neutrophils, and Dendritic cells in GBM.

Genomic alterations of DARS/GDI2/P4HA2/TRUB1 in GBM

We then used the cBioPortal tool to determine the types and frequency of DARS/GDI2/P4HA2/TRUB1 alterations based on DNA sequencing data from GBM patients. The genetic variation rates of DARS/GDI2/P4HA2/TRUB1 ranged from 0% to 4% (DARS was 4%, GDI2 was 1.4%, P4HA2 was 4%, TRUB1 was 0.0%). These alterations include Missense Mutation, mRNA High, mRNA Low, Amplification (AMP), and Deep Deletion (Fig. 7). In view of this, DARS and P4HA2 show potentially stronger cancer-driving properties at a higher mutation frequency. In contrast, TRUB1 is genetically stable and could potentially act as a stable biomarker.

Figure 7 Genomic alterations of DARS/GDI2/P4HA2/TRUB1 in GBM.

OncoPrint of DARS/GDI2/P4HA2/TRUB1 alterations in GBM cohort. The different types of genetic alterations are highlighted in different colors. Expression profiles of mRNAs showing different expression (≥3-fold) compared to control were considered to be mRNA high, and vice versa for low.

Gene functional analysis

We downloaded the DARS/GDI2/P4HA2/TRUB1 gene sets from the Molecular signatures database (v7.0 version) and comprehensively evaluated the gene sets through GSVA. Our analysis showed that in the DARS gene set, 17 gene sets were up-regulated (t>, 1) and 14 gene sets were down-regulated (t<1). In GDI2, 13 gene sets were up-regulated and 21 gene sets were down-regulated. In P4HA2, 11 gene sets were up-regulated and 30 gene sets were down-regulated. In TRUB1, 21 gene sets were down-regulated and 14 gene sets were down-regulated (Figs. 8A–8D).

Figure 8 GSVA analysis.

GSVA of DARS/GDI2/P4HA2/TRUB1 gene sets in GBM. (A) DARS. (B) GDI2. (C) P4HA2. (D) TRUB1.A t value > 1 or < − 1 represents statistically significant changes.

Discussion

Due to the complex mechanisms of GBM, it is one of the most threatening CNS malignancies. Therefore, it is an urgent need to find biomarkers related to the occurrence and prognosis of GBM to reveal the possible pathogenesis or predict the prognosis of patients, and then develop personalized treatment plans for GBM patients. Based on gene sequencing technology, we have discovered some biological markers with predictive value for patients including GBM. However, the role of these markers are still limited. In order to better understand GBM, there is an urgent need to screen out more biomarkers to improve the efficacy of GBM treatment and prognosis.

GBM, as a highly heterogeneous tumor harboring multiple genetic alterations (Harter, Wilson & Karajannis, 2014), molecular heterogeneity affects the effectiveness of single-molecule markers in predicting prognosis (Tonry, 2020). At the same time, some studies have found that the high recurrence rate of GBM is related to the expression of strong proliferation genes of cells (Lara-Velazquez et al., 2020). And these processes usually involve multiple genes (Malik et al., 2020). Therefore, we believe that multi-gene markers have a higher predictive power for GBM prognosis than single-gene marker. We built a multi-gene markers model for predicting GBM prognosis, and validated the multi-gene markers model through strategies including training, testing, and independent cross-validation. The above strategies significantly improve the predictive ability of genetic markers (Li et al., 2019b).

In our research and analysis, the results of GO and KEGG analysis indicate that cell transition metal ion homeostasis, protein hydroxylation, intrinsic apoptotic signaling pathway and other processes may play an important role in GBM. Among them, transition metals are critical for many metabolic processes (Nelson, 2014), and their steady state is vital to life. Aberrations in the cellular metal ion concentrations may lead to cell death and severe diseases such as cancer (Pi, Wendel & Helmann, 2020). Hydroxylation is a post-translational modification affecting protein stability, activity or interactome (Zurlo & Zhang, 2020). Many cancers are related to protein hydroxylation, such as breast cancer (Zurlo & Zhang, 2020), gastric cancer (Li et al., 2020), and prostate cancer (Della-Flora et al., 2020). For example, a study found that a set of enzymes PLOD1, PLOD2 and PLOD3 involved in the hydroxylation of lysine and stabilization of collagen by crosslinks, which up-regulated expression in gastric cancer patients (Li et al., 2020). Similarly, intrinsic apoptotic signaling pathway can activate or inactivate multiple signaling pathways and inhibit multiple tumor suppressor genes, thereby promoting tumor progression. Almost all cancers involve intrinsic apoptotic signaling pathway, including renal cell carcinoma (Chae et al., 2020) and multiple myeloma (Chen et al., 2020a). Combined with the above results, we believe that DARS/GDI2/P4HA2 / TRUB1 may be involved in these processes to affect the occurrence and development of GBM disease, which is also consistent with our Drug sensitivity analysis results. Among them, the DARS gene encodes the aspartyl-tRNA synthetase (Dominik et al., 2018), which pairs aspartate with its corresponding tRNA. Missense mutations in the gene encoding DARS can lead to leukocyte dystrophy, accompanied by a marked reduction in myelin sheath, abnormal movement and cognitive impairment (Fröhlich et al., 2018). There are no related reports about the relations between DARS and GBM. According to our research, DARS may participate in TSC/mTOR signaling, by regulating GBM cell growth process. GDI2 controls the activity of Rho GTPase’s pathway to regulatory guanine nucleotide exchange factor and GTPase activating protein, and may play a role in tumor cell apoptosis. This is also in line with our results. At the same time, a recent study shows that RhoGDI2 suppresses lung metastasis in mice by reducing tumor versican expression and macrophage infiltration. The expression of P4HA2 increased in head and neck squamous cell carcinoma (HNSCC) (Kisoda et al., 2020), Oral Squamous Cell Carcinoma (OSCC) (Reis et al., 2020), cervical cancer (Cao et al., 2020b) and other cancers. Especially, we found that P4HA2 are markedly upregulated in cervical cancer tissues and upregulation of P4HA2 was associated with shorter overall survival (OS) and relapse-free survival (RFS) (Cao et al., 2020b). In GBM, we found that P4HA2 is mainly involved in the process of inhibiting DNA damage, and is also related to EMT, Hormone AR, Hormone ER, RAS/MAPK, TSC/mTOR and other pathways. TRUB1 mRNA is widely expressed in various human tissues (especially heart, skeletal muscle and liver), but there are few studies on its relationship with cancer (Zucchini et al., 2003). In our research, we analyzed that TRUB1 is mainly involved in Apoptosis, DNA damage, EMT, PI3K/AKT and other processes.

In the analysis of key genes co-expression, we found the four genes (CDC73/CDC123/B4GALT1/CUL2) are most relevant to the expression of key genes and also related to the occurrence of many cancers. For example, CDC73 is a tumor suppressor, which can prevent cells from growing and dividing too fast or uncontrolled, and is closely related to parathyroid carcinoma (Cetani et al., 2019). CDC123 is a cell division cycle protein, and the regulatory effects of the entire cell cycle process can be stopped in one of the normal stages (G1, S, G2, M).CDC123 is highly expressed in choriocarcinoma (Hussain et al., 2018). B4GALT1 is one of the seven β-1,4-galactosyltransferase (beta4galt) genes. The β1,4-galactosylation of glycans is very important for many biological events, including the development of cancer. In a variety of cancers, the B4GALTs family is associated with cancer cell proliferation, invasion, metastasis, and drug resistance.B4GALT1 is highly expressed in patients with lung adenocarcinoma (Zhang, Zhang & Yu, 2019). CUL2is one of the seven members of Cullin family. It can participate in the regulation of cell cycle, proliferation, apoptosis, differentiation, gene expression, transcription regulation, signal transmission, damage repair, inflammation and immunity.CUL2 affects the occurrence of renal cell carcinoma by promoting the substrate ubiquitination and degradation (Liu, Zurlo & Zhang, 2020).

Further TIMER analysis indicated that the immune system had a good effect on tumor microenvironment, and that the mutations of DARS / GDI2/P4HA2/TRUB1 had important application value in tumor immunology. Finally, we conducted a comprehensive evaluation of gene sets using GSVA and we found that the DARS/GDI2/P4HA2/TRUB1 gene sets are closely related to the occurrence of cancer. For instance, the APICAL_ JUNCTION in the DARS gene set is more common in highly differentiated epithelial cells, such as colon cancer cells (Nair-Menon et al., 2020). MITOTIC_SPINDLE in the GDI2 gene set, the mitotic spindle inhibitor is one of the most commonly used chemotherapeutics now (Bukowski, Kciuk & Kontek, 2020). DNA_REPAIR in the P4HA2 gene set and ANGIOGENESIS in the TRUB1 gene set are also two important mechanisms of cancer development.

In recent years, with the GBM genes related to the occurrence and prognosis of feature recognition in many studies. Such as Chen X found the ASPMexpression pattern from the database showed that it is highly expressed in GBM tissue, and patients with high expression of ASPM have a poor prognosis (Chen et al., 2020b). Recently, a bioinformatic analysis of 123 GBM patients has established a 14-mRNA prognostic signature, which could be used to classify GBM patients into low and high risk groups (Arimappamagan et al., 2013). To our knowledge, the DARS/GDI2/P4HA2/TRUB1 that we identified are new GBM biomarkers because they have never been reported to be associated with the development and progression of GBM (Lu et al., 2020). At the same time, compared with the traditional typing methods, the multi-gene markers model has many advantages, such as high prediction accuracy and personalized detection results (Albuquerque et al., 2012). Therefore, multi-gene markers have a good application prospect in clinical practice. In our study, we built and verified the characteristic of the four genes through analyzing the two independent data sets. More reasonable use of biometrics and multiple independent data sets of mutual verification makes our results more reliable.

However, our study had some limitations. Associated with disease, for example, age, race, sex, and some unknown prognostic factors may not be included in the model, which limits the prediction ability of the model. In the future, we plan to establish a more reasonable model of biological information analysis. Meanwhile, it should be acknowledged that the single gene analysis in this study does have limitations, and in future studies we will combine all the key genes or other factors together to find a biomarker with better sensitivity and accuracy using a multi-omics approach. In summary, our results had shown that DARS/GDI2/P4HA2/TRUB1 can be used as a new biological marker for GBM, which is related to the occurrence and prognosis of GBM, how to rationally apply various genetic characteristics at specific stages of GBM for diagnose and prediction of prognosis.

Conclusion

The molecular biological characteristics of GBM has changed the classification and treatment of tumors and become an important part of diagnosis and oncologic therapy. This study used public databases to identify four valuable biomarkers for GBM prognosis, namely DARS/GDI2/P4HA2/TRUB1, which have potential and clinical application values to act as prognostic markers of GBM.

Supplemental Information

Supplemental Information 1 Specimen IDs acquired from the TCGA database

Click here for additional data file.

Supplemental Information 2 139 genes in the darkturquoise module

Click here for additional data file.

Supplemental Information 3 TCGA ID

Click here for additional data file.

Additional Information and Declarations

Competing Interests

Author Contributions

Data Availability

The authors declare there are no competing interests.

Mengyuan Zhang conceived and designed the experiments, performed the experiments, analyzed the data, prepared figures and/or tables, authored or reviewed drafts of the paper, and approved the final draft.

Zhike Zhou, Zhouyang Liu and Fangxi Liu analyzed the data, authored or reviewed drafts of the paper, and approved the final draft.

Chuansheng Zhao conceived and designed the experiments, authored or reviewed drafts of the paper, and approved the final draft.

The following information was supplied regarding data availability:

The raw data (TCGA search terms/accession numbers) are available in the Supplementary File.

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
