# Peer review of "Exploring the potential biomarkers for prognosis of glioblastoma via weighted gene co-expression network analysis"

_PeerJ, doi:10.7717/peerj.12768_

## Round 0.1 · original submission · Major Revisions

The reviewers have raised a number of important points. Please address them before resubmission.

·

Basic reporting

In the present study, the authors are intended to identify biomarkers that can be used for predicting the prognosis of glioblastoma. Overall, the study was well conducted, and the manuscript was clearly written. However, there are still some major/minor concerns.

1. The study is intended to identify biomarkers for GBM prognosis but not therapeutic targets, so I think the title of the manuscript is accurate and should be revised.

2. Please add the reference for the following sentences:
Introduction
(1) “Glioblastoma (GBM) is the most common primary neurogenic tumor, and the prognosis of most subtypes is poor.”
(2) “The main reasons for the poor prognosis of GBM are due to tumor metastasis and postoperative recurrence.”
(3) “Given that tumors invade the brain aggressively, GBM tumors can rarely be completely removed by surgery.”
(4) “In the past few decades, gene sequencing and bioinformatics analysis have been widely used for genetic variation screening at the gene level…”
Results
(1) “Among them, CDC73 and CDC123 are cyclins of cell division, B4GALT1 is one of …”
(2) “These four genes have been shown to have a close relationship with many kinds of cancers, such as thyroid carcinoma (Sarquis et al. 2019), breast cancer, etc”
Discussion
(1) “…molecular heterogeneity affects the effectiveness of single-molecule markers in predicting …”
(2) “At the same time, some studies have found that the high recurrence rate of GBM is …”
(3) “And these processes usually involve multiple genes.”
(4) “Among them, transition metals are critical for many metabolic processes, and their steady state is vital to life.”
(5) “…DARS gene encodes the aspartyl-tRNA synthetase,which pairs aspartate with its corresponding …”
(6) “… never been reported to be associated with the development and progression of GBM.”
(7) “… advantages, such as high prediction accuracy and personalized detection results.”

3. Please indicate why some samples are highlighted (green color) in figure 1a.

4. The resolution of some figures is low and the legends in those figures cannot be seen clearly.

5. Some abbreviations do not have full names (e.g. SCNAs, CNV). Please provide their full names.

6. As the authors described at the beginning of the manuscript, single gene analysis may not best predict the pathogenesis and prognosis of glioblastoma. Why not the authors establish a prediction model by combining all the hub genes and/or other factors together?

Experimental design

1. I am confused about the research goals of this study since the authors described that they want to identify biomarkers for glioblastoma while they compared the co-expressed gene modules between untreated glioblastoma patients and the recurrent cancer patients. I think a better experiment for knowing the biomarkers is to compare glioblastoma tissue with normal brain glia cells. If the authors want to know what genes are related to the recurrence of glioblastoma, a better comparison would be the recurrent cancer patients vs. non-recurrent cancer patients. Please explain why comparing untreated cancer patients with recurrent patients.

2. Please list all specimen IDs acquired from the TCGA database in a supplementary table.

3. Please clearly describe the criteria for choosing the four hub genes (DARS/GDI2/P4HA2/TRUB1) in the methods.

4. Please define “mRNA high” and “mRNA low” in figure 7.

Validity of the findings

1. The findings in figure 6a is not fully described in the context (e.g. P4HA2 and CD8+ T cells are also highly correlated). Please clearly describe all findings in the results.

2. Figure 6B shows that different mutations in the 4 hub genes can significantly affect immune infiltration, but not many genetic alterations were found in the GBM samples that are used in the study. Please explain this difference.

Reviewer 2 ·

Basic reporting

Zhang et al. used co-expression to identify several genes that are associated with the relapse/survival of glioblastoma. They subsequently investigated how these candidate genes were associated with known cancer-related pathways, drug response, immune response etc. While this study is well motivated, there are a few issues in how the authors designed the analyses and validated their findings.

Experimental design

See comments to authors.

Validity of the findings

See comments to authors.

Additional comments

1. The authors used GSE145128 to identify candidate genes. Since this is not a particularly big dataset, the generality of the authors’ findings is a concern. In addition, the samples of glioblastoma relapse in this dataset were all treated by temozolomide, raising the possibility that the markers identified by the authors can be related to this treatment. Unless the authors have a valid reason to use this single dataset (which they did not explain in the manuscript), I am not convinced that their findings generally apply to glioblastoma cases.
2. How the four hub genes were identified is not clearly described. Table 1 shows that there are 14 genes that are significantly associated with patient survival – how did the authors filter these 14 genes down to four candidates? Did the authors analyze GSE145128 to find the 14 genes, and then validate them with TCGA to find the four candidates? Since this part is unclear, I am not sure if the authors have tested whether their markers can predict patient survival, which undermines the validity of their central findings.
4. While the authors claimed that they were searching for “hub” genes in the darkturquoise module, they did not use the connectivity of a gene as the searching criteria. Instead, they used the correlation between genes and patient survival. To me, what they found should not be called hub genes, and the authors should rephrase the related texts.
5. I am not sure why the authors searched TCGA for genes that co-expressed with the four candidate genes. The rationale of WGCNA is clustering genes based on how well they co-express with each other, therefore the additional genes found in TCGA should already present in the darkturquoise module. On the other hand, if these genes were not found by WGCNA already, then the authors need to explain why they ignored the lack of the co-expression in GSE145128 and focused on TCGA.
6. The authors need to be careful with the analysis on immune cell filtration. Despite recent studies have suggested that certain T cells can populate brain, there may not be enough evidences that support the presence and functions of other immune cells in brain (I can very well be wrong on this point since I am not an expert in immunology). Can authors provide literature to support the presence and functions of B cells and neutrophils in brain?
7. I found the following figure legends unclear:
Fig2C: is each node a gene? What does the size of the node mean? The texts “Enrich the term network” are unclear.
Fig2B: the Y axis label “Gene significance for luminal” needs more explanation.
Fig3: what is the cutoff for low versus high expression?
Fig5A: what does the size of an area in the pie chart mean?

Minor:
1. The sections “Gene functional analysis” and “Cancer-related pathways and drug sensitivity analysis” are under the same topic. Perhaps the authors should reorder the sections?
2. The authors used “darkturquoise” to refer the WGCNA module of interest, but the term itself does not have a biological meaning, making it especially confusing when placed in the abstract. The authors should find a better term.
3. I think Fig1 should be moved to the supplement; it shows only technical details of the WGCNA analysis. Same for the most panels of Fig4, as they are showing redundant information.
4. The section “Genomic alterations of DARS/GDI2/…” is a bit thin. Can the authors say more on this topic, like are these mutations common enough to be used as therapeutic markers in GBM?

---

## Round 0.2 · Minor Revisions

In this round of the revised manuscript, questions raised by one of the reviewers were not addressed thoroughly. I think most the questions were related to the method details and interpretation of the related results. Please read the second reviewer's suggestions, and address his concerns seriously before the next round submission.

·

Basic reporting

All my questions were resolved satisfactorily by the authors. I suggest accepting the manuscript for publication.

Experimental design

N/A

Validity of the findings

N/A

Additional comments

N/A

Reviewer 2 ·

Basic reporting

I thank the authors for revising the manuscript, but some of my concerns were not adequately addressed. I am willing to consider the manuscript again, if the authors can address my questions seriously.

Experimental design

Please see comments to the authors.

Validity of the findings

Please see comments to the authors.

Additional comments

1. How the four hub genes were identified is still unclear, despite the authors’ explanation in the response. The authors wrote that “our primary criterion for identifying core genes was whether they had an impact on tumour prognosis, followed by KM survival analysis of all genes using the DARK module, in which DARS/GDI2/P4HA2/TRUB1 had a strong impact on survival and ranked higher.” Table 1 shows that genes like FKBP1B and DREU1 have lower P-value than DARS and TRUB1, then why aren’t these genes considered candidate genes? How was the strength of the impact on survival measured and ranked?
In addition, I had difficulty identifying the relevant changes in the manuscript, because there are a lot changes in the manuscript. Can the authors include a copy of the revised texts in their response?

2. To identify potentially important genes from a WGCNA module, both the connectivity of the genes and the associations between the genes and patient survival are valid criteria. But when the authors wrote that “hub genes have the highest connectivity in the module” and did nothing related to connectivity, it is really misleading. In their response to my comment, the authors claimed that the way they determined hub genes was identical to Lin et al. (2019). However, Lin et al. (2019) clearly wrote that “The hub genes of each module were selected based on the number of edges (degree of connectivity) associated with the corresponding nodes present within the network.”
I think “key gene” is a better term. Whether genes that are associated with survival necessarily have high connectivity is irrelevant to the goal of this study.

3. I asked authors to explain why they included additional genes that co-express with the four hub genes in the TCGA dataset. In their response, the authors cited Wang et al (2021) and Zheng et al (2021) to support their use of TCGA. However, these two studies did not use co-expression to identify the candidate genes from the non-TCGA data, so looking for additional genes that co-expressed with the candidate genes in TCGA does not produce contradiction. The current study identified candidate genes based on co-expression. If the expression of the candidate genes and that of the additional genes correlate positively in TCGA but negatively in the non-TCGA data, there will be a contradiction. While I understand it can be impractical to replace this analysis with a new one, the authors did not even try to address this problem in a meaningful way. With minimum effort, the authors could have verified if the expression of the additional genes and the four candidate genes are positively correlated in the non-TCGA data, or better verified if the additional genes are in the same or nearby WGCNA modules.

4. The authors’ response to my question on Fig2B is unfortunately not helping. Please define how “Gene significance for luminal” and “Module membership in darkturqouise module” in Fig2B were calculated.

---

## Round 0.3 · Minor Revisions

The current draft addressed most of both reviewers' concerns, please check the questions raised by reviewer 2 this round.

Reviewer 2 ·

Basic reporting

I thank the authors for clarifying my confusions! I can recommend the paper for acceptation, pending on a few minor revisions:

1. The authors have added texts to explain how the four candidate genes were chosen. Some references are needed after these sentences: Line 174 (PDF) "previous research experience in our lab" and Line 176 "confirmed to be related to tumorigenesis".

2. I am glad the authors found most of the co-expressed genes are also present in the darkturquoise module. To prevent other readers from being confused like I was, I suggest the authors notify readers about the membership of these genes in the co-expression analysis. Line 184, after "TRUB1 and CUL2 have the highest correlation", seems a good place for the notice and a reference to supplement 2.

3. Regarding Fig2B, the authors should explain the the x- and y-axis labels in the legend as they did in the response to my comment. Not all readers are familiar with WGCNA. I think explaining "module membership" as "correlation in expression between the given gene with the eigengene of the module" is sufficient. "Significance to luminal" seems to be a correlation between the expression of the gene and the trait luminal. The key question is how did the authors quantify this trait? Did they calculate it from certain information of the patients? Please clarifying this in the Materials and Methods section.

I trust the authors will fix these minor points, and I do not need to see a formal response again.

Experimental design

NA

Validity of the findings

NA

Additional comments

NA

---

## Round 0.4 · Minor Revisions

Most reviewer's questions are well addressed. However, some figures and fonts in them are still not clear enough to read. Please make quick adjustments in the next submission.

1. Fig1A, sample names too small.

2. Fig2A, Module/Trait table, p-values and correlation coefficients are too small to read. Since you only have 2 traits, you do not need that wide grid for one trait. The authors could extract the related values(p and R) from WGCNA, and plot similar figures themself if necessary. p-value and R could be put in the same line rather than 2 lines.

3. Fig4 A-D, enlarge the gene names. E-H, enlarge the gene names and figure legends. For A-D, if the legends were the same standard, you could just use one on the most right side. For E-H, the same suggestions for the fig legends.

4. Fig4 I-L, enlarge fig subtitles, axis titles. For the expressions, please note whether they were lcpm or tpm or variance stabilized transformation, or whatever you chose, in the figure legends.

5. Fig5A, enlarge the pathway names, align them with the pie charts. Enlarge the gene names.

Thanks for your time and Best regards,

---

## Round 0.5 · accepted · Accept

Congratulations! Your draft is now accepted for publication. Thanks for your effort.